# Mental Health Screening for Korean Ukrainian Refugee Minors in the Republic of Korea: A Cross Sectional Pilot Study

**Sejeong Park [1], Jenny Seongryung Lee [2], Hye-Jung Kim [3], Hojung Lee [2], Myungjoo Lee [2], Soo-Yeon Kim [2] and Han Choi [1,2,***

[1]   Graduate School of Art Therapy, Cha University, Seongnam 13488, Republic of Korea
[2]   Department of Medicine, Graduate School, Cha University, Seongnam 13488, Republic of Korea
[3]   A Psychological Development Clinic, Department of COA Otolaryngology, Incheon 21614, Republic of Korea
*   Correspondence: arttherapyhan@cha.ac.kr; Tel.: +82-31-881-7101

**Abstract:** Since February 2022, the Ukrainian refugee crisis has been highlighting mental health problems associated with trauma and distress. This study aimed to evaluate the mental health status of twenty-seven refugee minors (10 to 18 years old) who fled Ukraine and temporarily settled in the Republic of Korea (ROK). This cross-sectional survey study aimed to evaluate the mental health status of ethnic Korean Ukrainian refugee minors. The *Child and Adolescent Trauma Screen—Youth Report* (CATS), generalized anxiety disorder seven-item scale (GAD-7), and subjective unit of distress scale (SUDs) were used for assessment. A preliminary analysis indicated that 77% Ukrainian refugee minors were exposed to and experienced war-related trauma. They are at a relatively low risk of trauma symptoms, anxiety, and distress due to stable family and visa status and a comparably better environment in the ROK. Meanwhile, refugee teenagers showed higher rates of psychological distress compared with refugee children. This finding suggests that an early psychological interventions in a host country may be beneficial to prevent mental health issues in refugee minors.

**Keywords:** refugee; war in Ukraine; trauma; anxiety; depression; minor





## 1. Introduction

The escalation of full-scale conflict in Ukraine has caused the destruction of civilian infrastructure and civilian casualties, forcing people to flee their homes seeking safety and protection. Recent estimates suggest that 12 million Ukraine refugees have crossed borders into neighboring European countries [1]. Outside Europe, Ukrainians of Korean ancestry called "Koryo- saram" fled to the Republic of Korea (ROK). This refers to the descendants of ethnic Koreans who migrated to the former Soviet Union due to agricultural immigration, the anti-Japanese independence movement, forced mobilization, and other factors, from the 1860s to liberation, and those who are their relatives, currently residing in the region [2]. In response to the Ukrainian refugee crisis, the ROK government issued free entrance as humanitarian aid [3]. A study of internally displaced Ukrainians reported that approximately a third of Ukrainian refugees tend to meet the diagnostic criteria for post-traumatic stress disorder (PTSD) and anxiety [4–8].

A shortage of primary research regarding refugee crises may raise several mental healthcare issues in a host country [5–8]. Refugees who cannot return to their home country may be fearful of persecution in a third country. When a host country accepts refugees for resettlement, they are still vulnerable groups due to risks of poor mental health possibly influenced by past experiences. The rates of PTSD and depression are dramatically higher in refugee populations than those of the general population [4–8]. According to a comprehensive study on the prevalence of mental illness among refugees, 31.5% of refugees had PTSD, 31.5% had depressive disorders, and 11.1% had anxiety disorders [4]. Studies have reported that several factors affect the mental health of children and teenagers,

who currently comprise almost half of those fleeing the country [4,5]. First, during a pre-migration period, witnessing violent conflict is a crucial factor that affects refugees' mental health status. Secondly, during a peri-migration period, housing and employment are consistent elements related to increased mental health risks. Third, visa status, family reunification, environment, and age of arrival at the host country are important aspects to consider in the post-migration period [6–9].

During post-migration, refugees experience uncertainty about the outcome of their visa status, which can be a barrier to contacting healthcare services or pursuing job opportunities [9–11]; these barriers might be specifically related to the mental issues developed in refugees. For instance, in Australia, refugees with insecure visa statuses tend to have at least five times more anxiety symptoms than those with permanent residence [12]. Additionally, refugees with insecure visas showed significantly greater PTSD symptoms and had more suicidal intent than those with secure visas [12]. Visa insecurity is also related to resettlement issues, such as a lack of employment [13]. Refugees with visa security may allow more opportunities for refugees to obtain a stable job and build a social network. These opposites may provide a greater sense of social belonging [14,15]. Moreover, support for employment and acculturation from the government can help decrease depression and anxiety among refugees [16,17]. Refugees are often exposed to various stressful and traumatic experiences owing to forced separation from their families by deportation. Separated refugees often report more difficulties in living in a host country than non-separated refugees [18]. For example, refugees who spoke Arabic or Dari and were separated from family members reported a developed risk for mental disorders after arriving in their host country of southern Sweden [19]. Having family members in host countries is an effective mechanism for helping refugees adapt to a new society [19,20]. Specifically, Arabic-speaking refugees who arrive in European countries without a marital partner or a child tend to have more mental distress than those with all family members in a host country [20]. In Australia, elevated psychosocial problems were found among children with insecure residency and unstable family relationships compared to children with parents in secure residency [18,20]. This phenomenon demonstrates why family unification in a host country is fundamental to mitigating the risk of mental illness. Family reunion after arriving seems to be a crucial factor in promoting the mental health of young refugees [19,20].

In the life of refugees, migration is related to a variety of mental risks, which might differ depending on the social and economic status of refugees in their home country [21]. Specifically, refugees from the least developed parts of Syria were less affected by migration; instead, they seemed to benefit more from the current conditions as refugees in developed host countries. However, refugees who had a higher socioeconomic status in Syria were more likely to have a detrimental mental health status after migration. Furthermore, the atmosphere of the host country may affect the mental health of refugees [22]. Previous studies demonstrated that natives with low socioeconomic status feel more threatened by the presence of new migrants and are more likely to show discriminative attitudes [21,22]. For instance, some natives tend to have a pessimistic attitude toward new refugees [22]. Distinctively, low-middle-income countries host 74% of refugees worldwide [23]; however, these countries still have a relatively high level of poverty and social competition [21,22]. Some natives tend to believe that accepting refugees may increase economic competition because of the scarce resources or reduced employment opportunities [22–24]. Negative perceptions toward refugees may influence internalizing and externalizing behaviors, such as social withdrawal and criminal behavior [24]. This evidence shows that a lack of permissive attitudes in low-middle-income countries can be a threat to the mental health status of refugees [21,22,24,25]. However, if a host country can offer early healthcare services from a government and show a more permissive attitude toward refugees, the stress of post-migration may be mitigated [21,22,24–26].

Mental vulnerability in adolescence is a crucial factor as well [25]. Adolescence is a developmentally sensitive time [27,28], and for that reason, minors aged 10 to 18 years who have experienced traumatic events (e.g., sexual abuse, torture, and injury

during the war) are more vulnerable to stress-related disorders. Environmental changes and isolation from parents can be risk factors for mental illness, especially in refugee minors [27,28]. Specifically, age is associated with PTSD symptoms [28–30]. Refugee minors were more likely to have elevated mental health issues than refugee children. Age at arrival and gender were analyzed as possible predictors of mental health in different refugee populations [28–30]. Young male refugees tend to have a better mental health status than older refugee women [30,31]. Meanwhile, there is a relative dearth of information concerning refugees minors' mental health, even though psychological intervention is fundamental for refugee children of school age [28,30,32]. Thus, future research should highlight recent challenges and opportunities for these refugees to move forward [32].

The present pilot study explores the hypothesis that Korean Ukrainian refugee minors, aged 10 to 18, who fled to the ROK may experience psychological distress. The primary objective is to examine the levels of posttraumatic stress developed due to potentially traumatic experiences, severity of anxiety, and distress. The secondary objective is to compare posttraumatic-stress-associated mental health conditions between Korean Ukrainian refugee children and teenagers.

## 2. Materials and Methods

### 2.1. Study Design

This pilot study used a cross-sectional design. This study followed the strengthening the reporting of observational studies in epidemiology (STROBE) guidelines for cross-sectional studies [33,34].

### 2.2. Sample Size

Of the estimated 1600 Korean Ukrainian refugees who entered the ROK since Russia's full-scale military invasion of Ukraine in February 2022, an estimated proportion of 1.68% took part in the survey.

### 2.3. Setting

Data were collected from July to September 2022 in the two largest Koryo-saram settlements, Ansan City and Gwangju City of the ROK, closely associated with Koryo-saram communities and immigration support centers.

### 2.4. Particpants

Participants were recruited via a sampling frame of Ukrainians of Korean ancestry and their families. We chose Koryo-saram communities that have well established relationships with immigration support centers. The centers' collaborators identified and accessed potential participants by either screening pre-booked appointments or individual drop-ins. The inclusion criteria were as follows: (a) Refugee minors below the age of 18 years (grouping age criteria as determined by the United Nations and World Health Organization [35]); (b) Refugee minors who entered the ROK since the 2022 Russian invasion of Ukraine; (c) Refugee minors who willingly took part in the survey. Exclusion criteria included: (a) Refugee minors who were in psychotherapy during the study period. Participants were asked about their personal distressing experiences. Three measurements were used to assess the psychological distress of participants [36–43]. To prevent the risks of potential emotional disturbance, termination of the survey was conducted with participants who: (a) Withdrew consent; and (b) Had signs of emotional distress.

### 2.5. Assessments

Demographic information including age, gender, ethnicity, length of stay since arrival, and level of proficiency in the Korean language were collected.

### 2.5.1. Child and Adolescent Trauma Screen 7–17 Year Self-Check (CATS)

The child and adolescent trauma screen 7–17 years (CATS) [36] was used to screen for potentially traumatic events (PTEs) and posttraumatic stress symptoms following the DSM-5 [37]. The CATS was developed by a group of researchers internationally, and the researchers used Russian versions. First, the 15 PTEs checklist part was answered by yes or no. The checklist was based on the definition of trauma events in the DSM-5, which consists of natural disasters, accidents, violence, sexual abuse, traumatic loss, medical procedures, and war items in the home or community [37]. Without any reports on PTEs, further assessment was no longer required. If at least one was marked in the PTEs, the second part had to be checked. It consists of 20 items corresponding to the four symptom clusters in the DSM-5; criteria, B (intrusion) items 1–5, C (avoidance) items 6–7, D (negative alteration in cognition and mood) items 8–14, and E (hyperarousal) items 15–20. In the CATS, these four sub-scales are re-experiencing, avoidance, negative alterations in cognition and mood (NACM), and hyperarousal. The total score was calculated by summing up the raw scores of items 1–20; the possible range was 0–60. The Likert scale included four alternatives 0 = "Never", 1 = "Once in a while", 2 = "Half the time", and 3 = "Almost always". The total score interpretation was as follows: less than 15 was average, not clinically needed, 15–20 was considered moderate with trauma-related distress, and >21 indicated a clinically relevant level of symptoms, probable PTSD. Third, the five checklists assessed psychosocial functionality answers as yes or no. These five were about how the child or adolescent behaves around others, hobbies, school or work, family relations, and happiness [36,38]. CATS is license-free and can be used by clinicians and studies without permission [38].

### 2.5.2. General Anxiety Disorder-7 (GAD-7)

Anxiety disorder-7 (GAD-7) is a 7-item self-report scale that measures screening for generalized anxiety disorder and levels of anxiety severity [39–41]. This scale is based on the diagnostic criteria for GAD described in the DSM-IV [39]. The GAD-7 assesses GAD, panic disorder, social anxiety, and PTSD. These items asked participants to check the symptoms they experienced during the two weeks prior to answering the questionnaire. Each item was scored from 0 to 3 on a 4-point Likert scale: 0 = not at all, 1 = several days, 2 = more than half the days, and 3 = nearly every day. The addition of the scores of all seven items provided a total score ranging from 0 to 21. A total score of 5 or less was deemed "not at all", score of 6–10 was deemed 'mild', score of 11–15 was deemed "moderate", and score of 16–21 was deemed "severe" anxiety levels [40,41].

### 2.5.3. Subjective Unit of Distress Scale (SUDs)

The subjective unit of distress scale (SUDs) is a self-report scale used to measure recent psychological distress. This tool measures the intensity of feelings and other internal experiences, such as anxiety and agitation [42]. SUDs are simple analog scales that range from 0 to 100. The scale consists of one item and allows participants to check the intensity of subjective mental distress from perfectly relaxed (0) to the worst anxiety (100); A score of 0 is deemed "No distress; totally relaxed", a score of 10 is deemed "alert and awake; concentrating well, a score of 20 is deemed "minimal anxiety/distress", a score of 30 is deemed "mild anxiety/distress; no interference with functioning", a score of 40 is deemed 'mild-to-moderate anxiety or distress", a score of 50 is deemed "moderate anxiety/distress; uncomfortable but can continue to function". A score of 60 is deemed "moderate to strong anxiety or distress", a score of 70 is deemed "quite anxious/distressed, interfering with functioning". Physiological signs can also be observed. A score of 80 is deemed 'very anxious/distressed, cannot concentrate, and physiological signs are present. A score of 90 is deemed "extremely anxious and distressed". Lastly, a score of 100 is deemed the "highest anxiety/distress that a person has ever felt" [42,43].

### 2.6. Procedure

The sampling procedure is illustrated in Figure 1. Content validity was tested on Ukrainian and Korean bilinguals. Before starting the survey, its purpose and procedures were thoroughly explained by the researchers and interpreted by Ukrainian and Korean bilingual interpreters. Initially, 27 participants agreed to participate in a face-to-face survey. Written informed consent was obtained from all participants and their guardians. Two participants were excluded from the analysis because of incomplete questionnaires. A total of 25 participants completed the survey. Neither inter-rater reliability nor independent verification were performed. Participant data were recorded on a case report form. All the procedures were conducted and supervised by certified art psychotherapists.

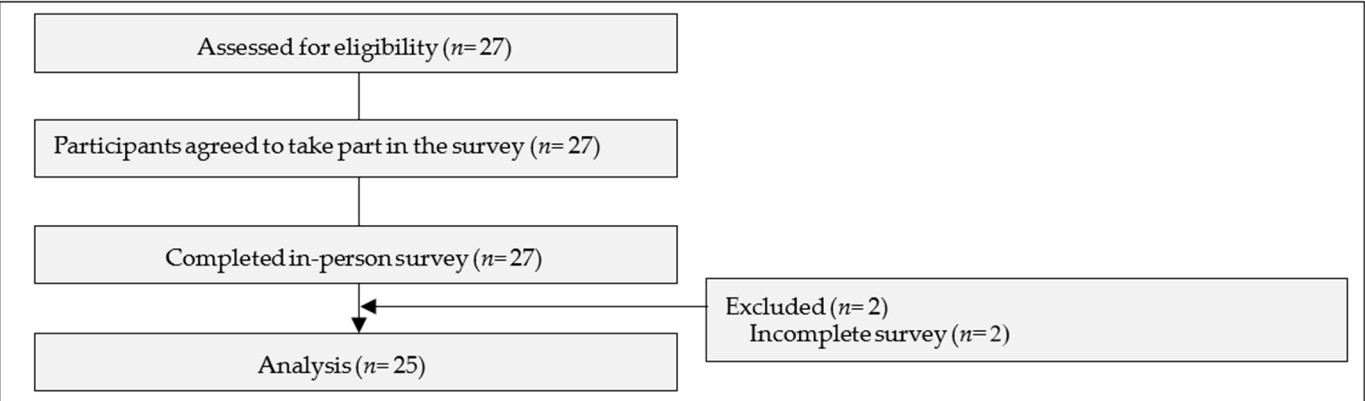

**Figure 1.** The sampling procedure.

### 2.7. Statistical Analyses

All statistical analyses were performed using SPSS Statistics 29.0 (IBM Corp., Armonk, NY, USA). Descriptive statistics were used to present the demographic characteristics of the sample, as well as the results of CATS (PTEs) and CATS (psychosocial functioning). An independent samples *t*-test was conducted between children and teenager groups to explore the differences in the scores on CATS, GAD-7, and SUDs. A *p*-value of less than 0.05 was considered statistically significant.

### 2.8. Ethical Considerations

This pilot study was approved by the Institutional Review Board of Cha University. All participants provided written informed consent before participating in the study. The purposes and procedures of the study were thoroughly explained to the participants, who were free to withdraw from the study at any time.

## 3. Results

### 3.1. Demographic Characteristics of Particpants

The demographic characteristics of the participants are presented in Table 1. The results of psychological measurements were analyzed using two sub-variables: age (children vs. teenagers) and gender (male vs. female). There were no significant differences between male and female. There was a significant difference between children (10–13 years old) and teenagers (14–18 years old). The criteria for age grouping were determined by the United Nations and the World Health Organization [31]. The mean age was 13.28 years.

### 3.2. The Screening Results

Table 2 shows the screening test psychological variables results. They show differences in mean scores for children and teenagers on CATS (PSS), GAD-7, and SUDs. Table 3 illustrates the screening results of psychological measurements (CATS: PTEs, CATS: psychosocial functioning) in the minor refugee population. The screening results of the CATS:

PTEs indicated that refugees reportedly experienced a variety of traumatic events. About 76% refugees reportedly experienced "being around war" and 44% reportedly experienced "seeing someone in your community get abused".

**Table 1.** Demographic characteristics of participants.

| Variables | Sub-Variables | *n* | % |
|---|---|---|---|
| Age | Children | 13 | 52 |
| | Teenagers | 12 | 48 |
| Gender | Male | 13 | 52 |
| | Female | 12 | 48 |
| Country of origin | Ukraine | 25 | 100 |
| Ethnicity | Korean | 25 | 100 |
| Length of stay since arrival | ≤ 1 month | 9 | 36 |
| | Between 2–6 months | 15 | 60 |
| | More than 6 months | 1 | 4 |
| Level of proficiency in Korean language | Low | 24 | 96 |
| | Moderate | 1 | 4 |
| | High | 0 | 0 |

**Table 2.** The screening test results of CATS (PSS), GAD-7 and SUDs.

| Variables | Sub-Variables | M(SD) | | |
|---|---|---|---|---|
| | | Children (*n* = 13) | Teenage (*n* = 12) | Adolescences (*n* = 25) |
| CATS (PSS) | Total | 1.78 (1.61) | 3.41 (3.17) | 2.56 (2.57) |
| | Re-experiencing | 0.49 (0.43) | 0.80 (0.65) | 0.64 (0.56) |
| | Avoidance | 0.57 (0.67) | 0.95 (1.03) | 0.76 (0.86) |
| | NACM | 0.37 (0.45) | 0.80 (0.79) | 0.58 (0.66) |
| | Hyperarousal | 0.34 (0.50) | 0.84 (0.82) | 0.58 (0.71) |
| GAD-7 | | 3.38 (3.64) | 5.25 (4.37) | 4.28 (4.03) |
| SUDs | | 20.91 (21.19) | 39.17 (23.91) | 30.43 (24.02) |

CATS = child and adolescent trauma screen 7–17 years self-check; PSS = posttraumatic stress symptoms; GAD-7 = anxiety-disorder-7; SUDs = subjective unit of distress scale; NACM = negative alteration in cognition and mood; M = mean; SD = standard deviation.

**Table 3.** The screening test results of CATS (PTEs), CATS (psychosocial functioning).

| Variables | Sub-Variables | Count (%) |
|---|---|---|
| CATS (PTEs) | Serious natural disaster | 3 (12%) |
| | Serious accident or injury | 4 (16%) |
| | Robbed by threat | 2 (8%) |
| | Abuse by family | 3 (12%) |
| | Abuse by someone | 6 (24%) |
| | Seeing someone in your family get abused | 3 (12%) |
| | Seeing someone in your community get abused | 11 (44%) |
| | Someone old touching your body | 1 (4%) |
| | Someone forcing sex | 0 (0%) |
| | Someone close to you dying suddenly or violently | 0 (0%) |
| | Attacked, stabbed, shot at or hurt badly | 0 (0%) |
| | Seeing someone attacked, stabbed, shot at, hurt badly or killed | 2 (8%) |
| | Stressful of medical procedure | 6 (24%) |
| | Being around war | 19 (76%) |
| | Other events | 4 (16%) |
| CATS (psychosocial functionality) | Getting along with others | 5 (20%) |
| | Hobbies/Fun | 6 (24%) |
| | School or work | 5 (20%) |
| | Family relationship | 2 (8%) |
| | General happiness | 8 (32%) |

CATS = child and adolescent trauma screen 7–17-year self-check; PTEs = potentially traumatic events, % = Percentage.

### 3.3. The Result of CATS between Two Population Groups

As illustrated in Table 4, the results of CATS: posttraumatic stress symptoms and CATS: psychosocial functioning between children and teenagers were not significantly different ($p > 0.05$). The results of the CATS (PTEs) indicated that children were more exposed to serious accidents and injuries than teenagers ($t = -2.30$, $p = 0.040$). Moreover, children were more exposed to "seeing someone in your community getting abused" ($t = 2.34$, $p = 0.028$) and "other events" ($t = -2.34$, $p = 0.040$) compared to teenagers.

**Table 4.** The result of CATS (PSS), CATS (PTEs), and CATS (psychosocial functioning) between two population groups.

| Variables | Sub-Variables | M(SD) | | *t* | *p* |
|---|---|---|---|---|---|
| | | **Children (*n*= 13)** | **Teenagers (*n*= 12)** | | |
| CATS (PSS) | Total | 1.78 (1.61) | 3.41 (3.17) | −1.59 | 0.130 |
| | Re-experiencing | 0.49 (0.43) | 0.80 (0.65) | −1.39 | 0.177 |
| | Avoidance | 0.57 (0.67) | 0.95 (1.03) | −1.10 | 0.281 |
| | NACM | 0.37 (0.45) | 0.80 (0.79) | −1.66 | 0.115 |
| | Hyperarousal | 0.34 (0.50) | 0.84 (0.82) | −1.84 | 0.078 |
| CATS (PTEs) | Serious natural disaster | 1.77 (0.43) | 0.00 (0.00) | −1.89 | 0.082 |
| | Serious accident or injury | 1.69 (0.48) | 0.00 (0.00) | −2.30 * | 0.040 |
| | Robbed by threat | 1.85 (0.37) | 0.00 (0.00) | −1.47 | 0.165 |
| | Abuse by family | 1.83 (0.38) | 1.92 (0.28) | −0.59 | 0.557 |
| | Abuse by someone | 1.77 (0.43) | 1.75 (0.45) | 0.10 | 0.915 |
| | Seeing someone in your family get abused | 1.92 (0.27) | 1.83 (0.38) | 0.66 | 0.511 |
| | Seeing someone in your community get abused | 1.77 (0.43) | 1.33 (0.49) | 2.34 * | 0.028 |
| | Someone old touching your body | 0.00 (0.00) | 1.92 (0.28) | 1.00 | 0.339 |
| | Someone forcing sex | 0.00 (0.00) | 0.00 (0.00) | - | - |
| | Someone close to you dying suddenly or violently | 0.00 (0.00) | 0.00 (0.00) | - | - |
| | Attacked, stabbed, shot at or hurt badly | 0.00 (0.00) | 0.00 (0.00) | - | - |
| | Seeing someone attacked, stabbed, shot at, hurt badly | 0.00 (0.00) | 1.83 (0.38) | 1.48 | 0.166 |
| | Stressful of medical procedure | 1.77 (0.43) | 1.75 (0.45) | 0.10 | 0.915 |
| | Being around war | 1.15 (0.37) | 1.33 (0.49) | −1.01 | 0.320 |
| | Other events | 1.69 (0.48) | 0.00 (0.00) | −2.30 * | 0.040 |
| CATS (psychosocial functionality) | Getting along with others | 1.69 (0.48) | 1.91 (0.30) | −1.34 | 0.194 |
| | Hobbies/Fun | 1.69 (0.48) | 1.82 (0.40) | −0.68 | 0.500 |
| | School or work | 1.69 (0.48) | 1.91 (0.30) | −1.34 | 0.194 |
| | Family relationship | 1.85 (0.37) | 0.00 (0.00) | −1.47 | 0.165 |
| | General happiness | 1.69 (0.48) | 1.64 (0.50) | 0.27 | 0.784 |

* PSS = posttraumatic stress symptoms, PTEs = potentially traumatic events, NACM = negative alteration in cognition and mood, M = mean, SD = standard deviation, * = $p = 0.05$.

### 3.4. The Result of GAD-7 and SUDs between Two Population Groups

As illustrated in Table 5, the results of the GAD-7 ($t = -1.16$, $p = 0.257$) and SUDs ($t = -1.93$, $p = 0.067$) between children and teenagers were not significantly different ($p > 0.05$).

**Table 5.** The result of GAD-7 and SUDs between two population groups.

| Variables | M(SD) | | *t* | *p* |
|---|---|---|---|---|
| | **Children (*n*= 13)** | **Teenagers (*n*= 12)** | | |
| GAD-7 | 3.38 (3.64) | 5.25 (4.37) | −1.16 | 0.257 |
| SUDs | 20.91 (21.19) | 39.17 (23.91) | −1.93 | 0.067 |

GAD-7 = anxiety disorder-7, SUDs = subjective unit of distress scale, M = mean, SD = standard deviation.

## 4. Discussion

This pilot study explored the impact of war-related psychological outcomes on Korean Ukrainian refugee minors who fled the ROK as a result of the 2022 Russian invasion of Ukraine. The results of this study suggest that refugee minors experienced traumatic events before arriving in the ROK and their psychological distress symptoms were not severe

enough to require screening for PTSD. In addition, we found differences between children's and teenagers' self-reports; the level of the CATS (PSS and psychosocial functioning), anxiety, and SUDs were higher among teenagers than among children, which is congruent with the results of previous studies [25,27–30]. These results indicate that teenage refugees may have experienced more severe and more distressing events than child refugees. However, only some PTEs results are statistically significant. The relatively low scores in exposure to potentially traumatic events and levels of anxiety severity suggest that there were some factors (age, family, visa, and economic development of the host country) that affected mental health stability in refugee minors.

### 4.1. The Effect of Family Relationships

Studies have suggested that refugee teenagers who experienced the war had significantly higher PTSD and anxiety scores than child refugees. However, the present study found that there were relatively low levels of posttraumatic stress symptoms among refugee minors, and that the positive effect of family relationships may affect refugees' mental health states [44–48]. This may have been due to distant family groups previously settled in the ROK [1,2]. This evidence indicates that if there is distant family in the refugees' host country, refugees' mental health states tend to be stable [44]. Refugees waiting for family reunification have been shown to feel distressed because of the uncertainty regarding the future. Uncertainty concerning the safety of their family adds cumulatively to stressors and aggravates mental health issues [44–47]. A previous study showed that refugees who did not have a family in the host country had a risk of mental disorders that was more than twice as high as those who had obtained family in the host country [44]. Refugee minors may experience severe symptoms of anxiety and PTSD in refugee camp [45]. However, they can overcome mental crises and are resilient to social and parental support. Factors that facilitate refugees' resilience include strong physical and mental support from their parents, families, and community [44,45]. These factors may reduce externalized or internalized mental problems. Furthermore, supportive families and communities can assist refugees in getting along with new host environments [46–48].

### 4.2. Stable Visa Status

The result demonstrates that relatively low levels of anxiety severity indicate that Korean Ukrainians may feel no urgency or pressure because of their stable visa status [49,50]. According to the ROK government, last March, the visa process has started for Korean Ukrainians, specifically with family members who reside in ROK [3]. Most of the participants in the study had legal permission to stay with their family members in the ROK; therefore, refugees do not have to worry about the visa process or deportation, which results in anxiety and depression [49,50]. Specifically, refugee minors who can live with their family members may not have any barriers to enrolling in a public school or visiting a social welfare center [48–50]. This permissive atmosphere may help refugee minors to recognize the host country as predictable and manageable [49], and previous research has suggested how restrictions on immigration affect migrants in the United Kingdom [50]. Undocumented migrants tend to have reduced trust and increased trauma and stress. They were at least five times more likely to report high depression and anxiety symptoms than those with a permanent visa [44,45]. Thus, undocumented refugees may experience more difficulty in connecting with society in the host country [47–50].

### 4.3. Economic Development of the Host Country

According to the United Nations, a high-income country means "those with incomes of more than $12,695" [51]. The ROK is ranked as a "high-income country"; however, Ukraine is ranked as a lower-middle-income country [51]. In 2022, the ROK will outstrip the OECD's average GDP as well, and by GDP, the ROK will be the 10th largest economy in the world. In contrast, Ukraine's GDP ranks 54 of 196 countries [52,53]. This study suggests that the developed environment may affect the mental health status of refugee minors [54].

For instance, refugees in developed countries tend to have fewer mental risks than those in developing countries; however, only a few studies have focused on the impact of the economy in a host country [54]. Developed countries can provide good labor markets for refugees' welfare and mental health status. Furthermore, when refugees settle in developed host countries, such as countries in Western Europe, they feel less resistance to integrate with a new host country that has fewer security issues [55]. Host countries' sufficient economic resources may provide proper economic assistance to refugee communities, and it also has a significant positive effect on refugees' living standards [54,55]. These previous findings also explain why Ukrainian refugees do not have severe anxiety and depression in the GAD-7 even though they experienced traumatic events; if refugees can receive appropriate legal and financial assistance from the host country, refugees can overcome their mental crises by themselves and engage easily in a new society [55,56].

### 4.4. Differences between Child and Teenage Refugees

Refugee teenagers scored higher than refugee children in the exposure to traumatic events category, specifically "serious accident and injury" and "seeing someone in your community got abused". This suggests that teenagers have developed insight and digested what happened around them while fleeing the ROK. Consistent exposure to accidents and injury means that individuals developed insight about damaging someone around them or their community as a result of the war [57]. Previous studies have also mentioned that these exposures can increase the risk of mental health problems for teenagers, especially during migration [58,59]. War experiences have been related to several psychological problems and symptoms among refugee teenagers [57–59]. Specifically, teenagers showed severe symptoms of anxiety, and post-traumatic stress [57]. Previous studies also insisted that age is an important socio-demographic variable in refugee research [59,60]. Fazel and colleagues [60] found a high prevalence rate of mental health problems among refugee teenagers. Statistics indicate that only between 13% and 21% of refugees seek mental health service; thus, urgent assistance and intervention policies for refugee minors are greatly needed [57]. Teenagers tend to endure long periods of violence and accidents during wars; therefore, they are likely to report more mental distress than children [61,62]. Previous studies also found that females reported more internalizing behavior problems than males [61–63]. However, males tend to show externalizing behavior problems [61,62]. However, this study did not find specific differences between males and females.

### 4.5. Limitations

This study suffers from a few limitations. First, surveying more participants could have increased the significance of the results. According to the statistical analyses of SUDs, the total scores rigorously support that refugee teenagers tend to feel more distressed than refugee children, but it is difficult to generalize due to the lack of participants. Therefore, future research may require a larger and more diverse sample size to increase results reliability. Second, this study used a face-to-face survey. Therefore, the participants' social desirability bias and a lack of objectivity should be considered [64]. Due to the sensitive content of the survey, participants might have felt uncomfortable revealing their emotions. Thus, further studies should consider research biases carefully and study how to mitigate the impact of social desirability bias (e.g., using an online survey format). Finally, we only analyzed refugee minors. Future studies should also consider other age groups and examine the differences between them.

## 5. Conclusions

The Ukrainian refugee crisis is an urgent global issue. Thus, this pilot cross-sectional study investigated the psychological distress of Korean Ukrainian refugee minors who fled to the ROK. The results suggested differences in psychological responses as consequences of war across refugee minors' age groups. Specifically, teenagers were more vulnerable to traumatic events. In addition, this study highlights that post-migration factors, such as

stable visa statuses and extended kin support by the host country, may influence mental health issues and make them less severe. A rigorous face-to-face survey and an increased sample size may facilitate the development of our understanding of mental healthcare for refugee minors seeking safety in the ROK.

**Author Contributions:** Conceptualization, H.C.; Methodology, H.C.; Data collection, J.S.L., H.-J.K., H.L., M.L. and S.-Y.K.; Formal analysis, S.P. and H.C.; Writing—original draft preparation, S.P., M.L. and H.C.; Writing—review and editing, S.P. and H.C.; Supervision, H.C. All authors have read and agreed to the published version of the manuscript.

**Funding:** This research received no external funding.

**Institutional Review Board Statement:** The study was conducted in accordance with the guidelines of the Declaration of Helsinki and approved by the Cha University Hospital Institutional Review Board (CHAIRB_BD_V.5.1_F011).

**Informed Consent Statement:** Informed consent was obtained from all the subjects involved in the study.

**Data Availability Statement:** The data supporting the findings of this study are available from the corresponding author upon reasonable request.

**Acknowledgments:** The authors are grateful to the Community Chest of Korea, a social welfare corporation, and the Neomeo nonprofit organization for supporting the Korean Ukraine refugee emergency support project. The authors would also like to thank the Cha University Multicultural/Diversity Art Therapy Team.

**Conflicts of Interest:** The authors declare no conflict of interest.

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
