# Peer review of "Mental Health Screening for Korean Ukrainian Refugee Minors in the Republic of Korea: A Cross Sectional Pilot Study"

_adolescents, doi:10.3390/adolescents3010011_

Round 1

Reviewer 1 Report

Background

It is not clear what the pre-test hypotheses are.

Methods

Was the study powered enough to make statistical inferences? I did not see any power calculations under “methods”.

Page 2, lines 74-75 vs lines 78-79 have contradictory statements regarding Syrian refugees from higher SES…lines 74-75 suggest that they have better mental health, but lines 78-79 state that they have worse mental health compared to refugees from poorer areas of Syria.

Page 4, line 184: “a score of 10” appears twice with different interpretations.

Discussion

Paragraphs 1-4 repeats content from “Results” and should be more condensed to allow more interpretation of already-stated results.

Page 9, line 298: Only “The” needs to be capitalized; this goes to all the titles under “Discussion”. Add an “-s” to “Relationship”.

Page 9, line 335: Take out the “s” at the end of “developments”. Add “the” in front of “host country”. This paragraph is interesting but needs a direct comment on Korea as the host country.

Second to last paragraph: the age difference is very interesting and one of the major findings of this study. It needs more references that explain possible developmental explanations behind this phenomenon.

Limitations: A major limitation is the lack of generalizability to other refugee contexts (different language, host country, ethnic background, etc.). The authors already mentioned the lack of power (see my earlier comment about the need for power calculations). Another limitation would be a lack of objective tools of measurement because all outcomes are self-reported. It is also a cross-sectional study without any longitudinal follow up. Mental health symptoms that show up later are not captured in this type of study.

Conclusion

There is a lot of repetition from “Results”. This section should be more concise about what are the key implications of the findings.

Author Response

Dear editior and reviewer,

I really appreciate your valuable comments. Please see the attachment below. 

Thank you so much.

Best, 

Sejeong Park

Reviewer 2 Report

Dear Authors, 

I have read your manuscript with interest. From my point of view, the paper addresses a very timely and urgent topic. Overall, I appreciated the manuscript which is well written and used a coherent and adequate methodology. 

I hope the following minor suggestions and comments will be helpful in order to improve and enhance the manuscript even more.

-Introduction

I would appreciate if the authors might give to readers the state of art of the researches on Ukrainian refugees (adult as well as adolescents and children), since one of the strong point of the article is to cover a very timely topic, even though no scientific references were given about it. This should be also appropriate in order to compare their results with the evidences on this topic within the Discussion section. 

Line 39-40: The Authors wrote: "The rates of post-traumatic stress disorder (PTSD) and depression are dramatically higher than those in the general population [4]". This sentence appears independent form the others, since the authors did not specified that the rates of PTSD and depression are higher for forced migrants compared to general population. Please, revise the sentence. Overall, I would suggest to update the literature on this point, since the authors quoted a paper quite dated (2015), while there is an ongoing and more recent literature they can refer. 

Line 43-46. I would suggest to the authors to be much more specific on this point, since when they said that several factors might affect the mental health of refugees, I think the authors referred to the combination of pre-migration, peri-migration and post-migration variables. Precision on this point should be added, since between these factors, the authors chose to focus much more on the post-migration ones. 

Line 112-115. I think these lines are not necessary, since the authors organized the paper following the scientific style of any article. 

Materials and Methods

In my opinion, one of the weak point of the study is that it addressed mental health of both children and adolescents (even though authors spoke about adolescents aged between 10 and 18 yrs old, in literature we usually refer to adolescence from 11 yrs old). If the participants were all minors (< 18 yrs old), I would suggest to refer to and name them as "minors" (also within the title). If not, the authors should find a more appropriate label to name their participants.

The Materials and Methods as well as the Results section are, in my opinion, the most problematic ones.

Firstly, I think the authors should clarify where the participants were recruited, since they did not mention this issues at all. Moreover, in some part of the paper (both within text as well as within Tables) they referred to participants as children and teenagers, while in other parts, they spoke about participants dividing them in children, teenagers and adolescents. Strict age ranges should have been established from the authors before conducting the study. Consequently, these ranges should be well described  within the Participants section as well as maintained homogeneous through the paper. Due to the small number of participants, I think the authors should be as precise as possible on this issue. Otherwise, the reading might appear very chaotic and difficult. 

Line 231-232: "The screening results indicated that psychological measurements were higher in adolescents.". This sentence is incorrect. I believe the authors wanted to highlight that the levels of the psychological measurements were higher. Please, be more precise and clear in describing your results.

Sincerely,

Best regards

Author Response

(The authors gave the same response as above.)
